# Genotype-by-Diet Interactions for Larval Performance and Body Composition Traits in the Black Soldier Fly, *Hermetia illucens*

**DOI:** 10.3390/insects13050424

**Published:** 2022-04-30

**Authors:** Christoph Sandrock, Simon Leupi, Jens Wohlfahrt, Cengiz Kaya, Maike Heuel, Melissa Terranova, Wolf U. Blanckenhorn, Wilhelm Windisch, Michael Kreuzer, Florian Leiber

**Affiliations:** 1Department of Livestock Sciences, Research Institute of Organic Agriculture (FiBL), Ackerstrasse 113, 5070 Frick, Switzerland; si.leupi@gmail.com (S.L.); jens.wohlfahrt@fibl.org (J.W.); cengiz.kaya.vio@hotmail.com (C.K.); florian.leiber@fibl.org (F.L.); 2Institute of Agricultural Sciences, ETH Zurich, Eschikon 27, 8315 Lindau, Switzerland; maike.heuel@usys.ethz.ch (M.H.); michael.kreuzer@usys.ethz.ch (M.K.); 3Department of Evolutionary Biology and Environmental Sciences, University of Zurich, Winterthurerstrasse 190, 8057 Zurich, Switzerland; wolf.blanckenhorn@uzh.ch; 4AgroVet-Strickhof, ETH Zurich, Eschikon 27, 8315 Lindau, Switzerland; melissa-terranova@ethz.ch; 5Animal Nutrition, TUM School of Life Sciences, Technical University Munich, Liesel-Beckmann-Strasse 2, 85354 Freising-Weihenstephan, Germany; wilhelm.windisch@tum.de

**Keywords:** feeding value, genotype-by-environment interaction, genetic differentiation, insect-livestock, insect-microbiota, microsatellite markers, mitochondrial COI, nitrogen-to-protein conversion, phenotypic plasticity

## Abstract

**Simple Summary:**

The bioconversion of organic waste into valuable insect protein as an alternative animal feed ingredient has the potential to improve agricultural sustainability and may become a key element of future circular economy. However, while insects farmed for feed production are considered livestock from a regulatory perspective, systematic linking of genetic resource characterisations and fundamental phenotyping, crucial for precision breeding and feeding schemes, remains scarce even for prime insect candidates, such as the black soldier fly (BSF). The present study initiated to fill this knowledge gap by experimentally assessing BSF genotype-by-diet interactions for a number of economically and ecologically relevant larval phenotypic traits. Besides pervasive diet effects, strong impact of BSF genetic background and ubiquitous environment-mediated interactions were found. This implies some of the so-far unexplained response variation across global BSF studies could be driven by previously neglected mechanisms of genetic specificity, and thus that the concept of broad conspecific plasticity in this insect is likely too simplistic. Instead, it is emphasised that matching BSF genetics to dietary contexts is vital for purposive production optimisation, particularly when extrapolated to large-scale operations. These insights highlight that establishing tailored BSF breeding as an independent branch offers veritable opportunities to efficiently support this growing agricultural sector.

**Abstract:**

Further advancing black soldier fly (BSF) farming for waste valorisation and more sustainable global protein supplies critically depends on targeted exploitation of genotype-phenotype associations in this insect, comparable to conventional livestock. This study used a fully crossed factorial design of rearing larvae of four genetically distinct BSF strains (*F*_ST_: 0.11–0.35) on three nutritionally different diets (poultry feed, food waste, poultry manure) to investigate genotype-by-environment interactions. Phenotypic responses included larval growth dynamics over time, weight at harvest, mortality, biomass production with respective contents of ash, fat, and protein, including amino acid profiles, as well as bioconversion and nitrogen efficiency, reduction of dry matter and relevant fibre fractions, and dry matter loss (emissions). Virtually all larval performance and body composition traits were substantially influenced by diet but also characterised by ample BSF genetic variation and, most importantly, by pronounced interaction effects between the two. Across evaluated phenotypes, variable diet-dependent rankings and the lack of generally superior BSF strains indicate the involvement of trade-offs between traits, as their relationships may even change signs. Conflicting resource allocation in light of overall BSF fitness suggests anticipated breeding programs will require complex and differential selection strategies to account for pinpointed trait maximisation versus multi-purpose resilience.

## 1. Introduction

Insect farming for feed applications in aquaculture and monogastric livestock production is being established as a new agricultural sector [1,2,3,4]. In line with the concept of a circular economy, substituting conventional animal feed ingredients characterised by high ecological footprints with insect-based components obtained via recycling nutrients from organic waste may overall increase sustainability and food security alike [5,6,7,8,9,10]. Accordingly, not only from a regulatory perspective [5,11], mass-produced insects for feed can themselves be considered a novel livestock. The black soldier fly (BSF), *Hermetia illucens* (L. 1758; Diptera: Stratiomyidae), is the most prominent globally farmed insect and is readily amenable across diverse settings worldwide, ranging from a smallholder context to highly automated industrial scales [12,13].

Products based on BSF larvae (BSFL) are considered highly valuable and digestible ingredients for aquaculture [14,15,16,17], poultry [18,19], and pig [20,21] feeding. However, variation in nutritional profiles of BSFL may have important consequences on their feeding values [3,22]. Indeed, responses in layer and broiler performance varied when fed formulations containing two different brands of partly defatted BSFL meals and their separately incorporated lipid fractions [23,24,25]. Yet, the extent to which these effects may be attributed to different diets used for BSFL fattening or genetic differences between BSF strains remained unclear. Numerous studies rearing BSFL on different diets evidence that nutrition strongly affects commercially relevant life-history traits (e.g., [17,26,27,28,29,30,31,32]). Performance on seemingly comparable diets across different studies nevertheless documents notable variation [5,6,33]. The same is true for compositional characteristics: individual studies not only detected substantial variation in the proximate composition of BSFL, including crude protein, lipids, and total ash but also subtle yet highly important variability in amino acid and fatty acid profiles [34,35,36,37,38,39,40]. Similarly, while nutrition is indicated to be a major driver of variation in BSFL body composition [22], available reviews across settings worldwide further reveal that differences other than the developmental stage at harvest [41] may not solely be related to diets [6,22,42,43] but also involve other factors.

The potential role of BSF genetic background in influencing these patterns has so-far been virtually neglected in farmed insects [44], contrary to conventional farmed animals where detailed characterisations of genetic resources are systematically implemented in breeding programs. While variation in performance and body composition in relation to diet across populations has only recently been documented for the yellow mealworm [45], Zhou et al. already described such remarkable patterns for the BSF almost a decade ago [46]. However, these authors supplied no genetic characterisations of the used strains. Later studies limited to maternally inherited mitochondrial markers actually indicated considerable intraspecific genetic variation for the BSF [47,48]. In a recent article about standardisations in BSF research, it was, therefore, recommended to generally specify the used BSF origins (in addition to, e.g., diet specifications) to account for putative strain effects in future meta-analyses [49]. In the meantime, Kaya et al. [50] presented a robust microsatellite population genetic reference dataset documenting substantial nuclear genetic variation and population structure worldwide and ultimately permitting the straightforward assignment of BSF specimens to differentiated genetic clusters anywhere around the globe. Thereby genetically determined performance and body composition trait variation relative to diet-mediated effects can be disentangled. Such fundamental assessments are indispensable, both for evaluating the breeding potential of selected BSF traits [51] and exploring the genetic architecture of traits based on recently published genomic resources [52,53]. For instance, Kaya et al. [50] documented a shared ancestry across most BSF strains used for academic research and commercial purposes across North America, Europe, Africa, and Asia. Yet, distinct breed formation derived from this origin could still translate into trait variation, even among relatively closely related BSF lineages. Most importantly, however, this study identified highly structured and differentiated wild populations across the indigenous ranges of the Americas, as well as throughout non-native areas worldwide where the BSF became naturalised. Its present cosmopolitan distribution characterised by largely unique regional genetic profiles may not have solely been shaped by demographically-mediated processes of genetic drift but effectively reflect ecological adaptation as well [50].

This raises the question to which extent plasticity enabling the BSF to exploit different dietary resources may be driven by previously ignored effects of BSF genetic background and genotype-by-environment (G × E) interactions, i.e., the dependency of differential performance of given genotypes (either individuals or entire populations) on the actual environment [44,54]. Generally, depending on the environmental context, prominently including diet, strong G × E patterns have been observed in various insects, such as aphids, *Drosophila*, butterflies, and honeybees [55,56,57,58]. The presence and magnitude of G × E interactions, as indicated by non-parallel or even crossing reaction norms of genotypes across environments, is important for inferring the complexity of genetic architectures of traits in livestock in order to evaluate their heritability and potential for improved breeding. Furthermore, correlations of a given trait across environments, as well as the nature and strength of correlations between traits, are key for outlining appropriate breeding programs [54].

Our study is the first that systematically assesses G×E interactions in the BSF. We implemented a fully crossed factorial design by rearing four genetically distinct BSF strains on three qualitatively different diets. This work aimed at evaluating the effects of BSF strain relative to diet, as well as potential interactions between them for a number of larval performance and body composition traits. We anticipate that these detailed insights into fundamental biological patterns will trigger the re-thinking of application-oriented perspectives. They will moreover be valuable for refining future breeding goals for the BSF and should further support economic viability, as well as ecological sustainability of the sector.

## 2. Materials and Methods

### 2.1. Genetic Characterisation of Experimental BSF Strains

From a larger pool of BSF populations cultured at the Research Institute of Organic Agriculture, four strains (S1–4) sourced from four different continents were used for the experiment. The selection was based on diagnostically distinct nuclear genetic profiling at 15 microsatellite loci, as detailed by Kaya et al. [50], and complemented by mitochondrial haplotype profiling using partial cytochrome c oxidase I (COI) gene sequences following Ståhls et al. [48]. Specifically, referring to the characterisation of worldwide BSF genetic clusters of Kaya et al. [50], our study included one experimental strain each matching cluster 1 (i.e., S4; sourced from an African facility), and cluster 2 (i.e., S1; sourced from a European facility), respectively. Both these virtually exclusively captive clusters go back to the same North American origin, comprise the strains most widespread across commercial BSF operations worldwide, and exhibit pronounced differences from wild BSF populations, including genomic signatures of domestication [50]. The other two strains were identified as a characteristic representative of Asian naturalised wild populations of genetic cluster 6 (i.e., S2; sourced from Southeast Asia) and cluster 8 comprising naturalised eastern Australian populations (i.e., S3; sourced from eastern Australia), respectively (cf. [50]). Both these distinct genetic clusters were inferred to have been formed successively upon natural colonisation across non-native Australasia, there being highly abundant in regional BSF facilities [50]. All relevant strain information is summarised in Appendix A.

For documentation purposes, a representative sample of 50 larvae of each strain was randomly and evenly selected from all replicates across diets (i.e., two or three individuals per experimental unit, see below) at the end of the experiment to be subjected to DNA extraction and microsatellite genotyping. Basic population genetic characteristics, cluster analysis, pairwise differentiation between strains, and visualisation of genetic relationships using discriminant analyses of principal components (DAPC) were assessed using the R packages and procedures described in Kaya et al. [50] (see also reference therein). In brief, nuclear genetic analyses were computed using the R (v. 3.6.1) packages (https://www.R-project.org/) *adegenet* [59,60], *hierfstat* [61], *pegas* [62], *PopGenReport* [63,64], *poppr* [65,66], and *strataG* [67] to evaluate potentially matching multi-locus genotypes (MLG), number of total (*N*_A_) and unique (*A*_U_) alleles per locus and strain, allelic richness (*A*_R_) per locus and strain, *F*-statistics (*F*_ST_ and *F*_IS,_ [68]), including global and locus-/strain-specific tests, as well as observed and expected heterozygosity (*H*_obs_, *H*_exp_). Effective population sizes of the strains’ parental generation were calculated in *NeEstimator* [69] based on the linkage disequilibrium test and conservatively excluding unique alleles. The optimal number of genetic clusters (*K*) in the data was inferred using a maximum-likelihood procedure that combines a geometric approach with the Expectation-Maximisation algorithm based on Kullback Information Criterion (KIC) goodness-of-fit statistics [70,71], as implemented in *adegenet*. Model convergence was verified, and posterior membership probabilities of all individuals to different clusters were visualised by stacked bar plots using *ggplot2* [72]. Further, *adegenet* was used to plot individuals based on their multi-locus genotypes in a multidimensional space using DAPC, a multivariate method that focuses on variances between groups while minimising within-group variation to characterise population subdivision equivalently but faster than Bayesian algorithms [73], with retained principal components being cross-validated to avoid overfitting.

Mitochondrial COI sequences were generated from an average of 16 randomly chosen specimens per strain and revealed 4 haplotypes across strains: while S1 harboured 2 distantly related haplotypes, the other strains harboured different but single haplotypes exclusively, with S2 and S3 being fixed for the same haplotype. All strain-specific mitochondrial haplotypes matched expectations regarding biogeographic origins and breeding history, as detailed by Ståhls et al. [48]. Mitochondrial haplotype relatedness among strains was computed based on uncorrected p-distances and depicted using *MEGA* [74], as shown in Appendix A.

### 2.2. Rearing of Experimental BSF Strains

All strains were reared under identical controlled conditions for at least two full generations prior to the experiment to minimise putative maternal effects. To ensure all strains were naïve to the effective experimental diets, basic strain maintenance prior to the experiment included a fattening diet that was not part of this study (i.e., brewer’s spent grains). Large stock populations of each strain were maintained in custom-made walkable double-net-systems (L × W × H of inner net: 1.6 × 0.8 × 1.6 m) to prevent cross-contamination. For producing the parental generation of the experimental larvae, equal numbers of pupae per strain were transferred to insect cages (BugDorm-44590) placed strain-wise within walkable net cages equipped with 150W UV-light emitting LED panels (evo Conversion Systems LLC). Upon adult emergence, flies could mate and oviposit into cardboards placed on plastic cups equipped with an oviposition attractant. Only eggs laid within a 24-h period were collected for the experiment to ensure synchronisation, which were transferred to aerated, escape-proof plastic containers kept in a climate cabinet (KBF 720, Binder GmbH, Tuttlingen, Germany) at 28 °C and 60% humidity. The hatched neonates were allowed to feed on moistened (70%) compound feed for laying hens (UFA 625 Crumbs IPS, UFA AG, Herzogenbuchsee, Switzerland) ad libitum for 5 days before the experiment started (see below).

### 2.3. Experimental Diets

The three different experimental diets were poultry feed (PF; described above), pre-consumer food waste (FW), and poultry manure (PM). The FW consisted of green cabbage, whole bananas, and pre-cooked ‘Spätzle’ type pasta mixed in a 1:1:2 ratio (wet weight basis). Individual components were ground with a food waste recycling machine (WEREC-Spüler F-650, WEREC AG, Gundetswil, Switzerland) and mixed thoroughly to obtain a homogenous diet. Fresh PM was collected from a Swiss organic layer house. Absolute dry matter (DM) contents (g/kg) inferred by lyophilisation were similar across experimental diets (PF: 263; FW: 319; PM: 294). For all experimental diets, two-day rations containing feeding rate equivalents of 77 mg wet weight per larvae per day were frozen at −18 °C and subsequently thawed at 27 °C for 24 h prior to administration every other day after experimental start (day 0). These rations are well comparable to previous studies [26,34,43].

Dietary nutritional compositions clearly differed, as summarised in Table 1. Generally, the first prepupae occurred slightly earlier on PF than on FW and PM, as expected. Therefore, to ensure all treatments were harvested during the same developmental stage (see below) while preventing feed limitations, all strains were fed until day 12 (seven rations) on PF but received an eighth ration on day 14 on FW and PM. Consequently, diet-specific dry matter contents, together with the adjusted ration numbers, ultimately resulted in different total provisions per replicate across diets, which were, however, the same for all strains: 312 g for PF, 432 g for FW, and 399 g for PM.

### 2.4. Experimental Procedures

Neonates of each strain that hatched within a synchronised 12 h age cohort went through five days of nursery (see above) before they were separated from residues (frass) by sieving (1 mm mesh size), which also ensured larvae of homogenous size. Each of the twelve treatment combinations was replicated six times. Thus, we counted 18 times 1100 larvae individually per strain and group-weighed them before allocation to experimental plastic containers (L × W × H: 25 × 16.9 × 11.4 cm; Auer Packaging Swiss GmbH, Baar, Switzerland). The lids were equipped with fine mesh to allow gas exchange and prevent larvae from escaping. During the experiment, the containers were kept at 28 °C and 50% humidity in a climate cabinet (KBF 720, Binder GmbH, Tuttlingen, Germany). Moderate humidity was chosen to facilitate the separation of larvae from residual slurry upon later harvest. During the experiment, the containers were randomly placed on the racks and rearranged (levels plus position within levels) every two days after feeding to avoid any positional bias. Feeding was stopped for a given treatment combination when the first prepupae were observed, and the larvae were harvested one day later to ensure: (i) overall harmonised developmental stages to be later subjected to body composition analyses and (ii) that peak growth was achieved before ceasing feed uptake upon transition to the prepupal stage (6th larval instar).

### 2.5. Assessment of Larval Development and Sample Processing

After the start of the experiment with five-day-old BSFL (day 0), random samples of 30 larvae per container were collected on days 4, 8, and 12 to determine average larval weights. Larvae were rinsed, dried with paper towels, weighed, and returned to their container. Larvae were handled the same way after separation from residues at harvest before counting and weighing to assess mortality, total larval biomass, and average larval weights. Without prior starving (as common practice in production plants), the larvae were then sacrificed by freezing. Leftover residue biomass was calculated based on total container weight before separation minus total larval biomass and empty container weight. Representative samples of each diet and BSFL and residues of each replicate were stored at −18 °C prior to freeze-drying and milling through a 1 mm screen (Retsch Ultra Centrifugal Mill ZM 200, Retsch GmbH, Haan, Germany).

### 2.6. Compositional Analyses and Calculations

The proximate composition of diets, residues, and harvested BSFL (plus extra samples of initial 5-day-old BSFL) were analysed in duplicates using AOAC standard procedures [75]. The determination of dry matter (DM) and total ash (TA) was conducted with an automatic thermogravimetric determinator (TGA-701, Leco, St. Joseph, MI, USA; AOAC index no. 942.05). The ether extract (EE), an estimate of fats and other ether soluble compounds was determined using petroleum ether (Büchi Extraktionssystem B-811, Büchi Labortechnik AG, Flawil, Switzerland; AOAC index no. 963.15). The total nitrogen and carbon contents were measured with an automatic C/N analyser (TruMac CN, Leco, USA; AOAC index no. 968.06). We inferred the estimated protein (EP) contents of whole larvae based on 4.76 × N following Janssen et al. [76] to conservatively account for non-protein nitrogen (NPN) components, such as chitin. Contents of neutral detergent fibre (NDF) and acid detergent fibre (ADF) of diets and residues were determined following VDLUFA [77] methods 6.5.1 and 6.5.2, using heat-stable α-amylase for NDF analysis (Fibertherm FT 12, Gerhardt, Königswinter, Germany). The contents of NDF and ADF were corrected for residual ash, and hemicellulose (HC) was calculated as the difference between NDF and ADF. We inferred BSFL bioconversion efficiency (BE) on a DM basis following Bosch et al. [49] as
$$\frac{g\;BSFLharvest-g\;BSFLstart}{kg\;diet},$$
which we further extended accordingly to analytically determined nitrogen efficiency (N-BE). Similarly, we estimated reduction rates (RR) of DM, NDF, ADF, and HC as
$$1000\;g-\frac{g\;residue}{kg\;diet}$$

Moreover, we calculated DM losses from experimental containers based on DM mass balances as
$$1000\;g-\frac{g\;residue+g\;{BSFL}_{harvest}\;}{kg\;diet+kg\;{BSFL}_{start}}$$

The latter combines overall gaseous emissions related to the insects’ respiration and digestion plus various volatile compounds deriving from microbial decomposition.

The contents of individual amino acids (AA), except Tryptophan (*Trp*), were determined by ion-exchange chromatography [78] in duplicates for each diet and larval replicate within treatment combinations. Asparagine plus aspartate, as well as glutamine plus glutamate, were not differentiated analytically but combined as *Asx* and *Glx*, respectively. The summed AA contents, including proxies for *Trp*, allowed enumerating specific protein (SP) concentrations (g/kg DM). Accordingly, profiles of individual AA (g/100 g protein) for BSFL were calculated by accounting for a general literature-based proxy for *Trp* of 1.8 g/100 g protein in BSFL [25,36,37,38,76]. Similarly, a common constant of 1.2 g *Trp*/100 g protein was applied to all diets (e.g., [36,79]) to provide merely informative specific protein concentrations of different diets. Further, based on analysed nitrogen contents, specific nitrogen-to-protein conversion factors (*Kp*) were derived replicate-wise for BSFL and generally for diets to specifically correct for NPN, such as uric acid in poultry manure or chitin in insects.

### 2.7. Statistical Evaluations

All data were analysed and visualised using appropriate R (v. 3.6.1) packages (https://www.R-project.org/), such as *car* [80], *ggplot2* [72], *lme4* [81], *lmerTest* [82], *lsmeans* [83], *MASS* [84], *multcomp* [85], or *phia* [86]. 

Growth dynamics over time were analysed using a linear mixed-effect model fitted with the threefold interaction of the explanatory factors (strain, diet, day) as fixed effects and the replicate unit as a random effect. The global test assessing all main effects plus two and threefold interactions indicated that pairwise post-hoc contrasts between levels within a given factor across levels of the other factors were inappropriate. Therefore, because we were particularly interested in potential interactions of genotype and diet during individual growth phases, additional post-hoc contrasts of the threefold interactions were performed for successive assessments throughout the experiment. Moreover, pairwise contrasts between strains within specific levels of the factors diet and time were investigated in-depth.

All response variables omitting a temporal component were subjected to a two-factorial analysis of variance. Full models were fitted with strain and diet as factors plus their interaction. Absolute responses, including average BSFL weight at harvest and total amounts (g DM) of larval biomass, EP, EE, TA, as well as EP:EE ratios, were log-transformed to stabilise variances, while proportional data for the response variables mortality, BE and N-BE, and reduction rates of DM, NDF, ADF. and HC were arcsine square root-transformed accordingly. Different variances across combinations prompted us to use weighted modelling, with weights being set as the inverse of the residual variances of the response. Proportional data of BSFL concentrations (g/kg DM) of EP, EE, and TA, as well as estimated DM losses, were analysed using generalised linear models with log-link and binomial (TA) or quasi-binomial error distributions to account for overdispersion (EP, EE, and DM losses). The evaluations of individual amino acids (g/100 g protein), SP (g/kg DM), and *Kp* did not require data transformation and were selectively subjected to weighted linear models if indicated by model diagnostics (cf. above). In addition to significance testing of main effects and their interactions for each response variable, key drivers of significant interaction terms were assessed by pairwise post-hoc contrasts of the twofold interactions (18 tests across 4 strain and 3 diet levels). Upon significant interactions, pairwise post-hoc contrasts between strains were only computed within but not across diets. Since pronounced diet effects were a priori expected, no additional post-hoc contrasts between diets within strains were assessed, except for amino acid and SP concentrations and *Kp*. Alpha levels were adjusted for multiple testing throughout.

## 3. Results

### 3.1. Population Genetic Characteristics of Experimental BSF Strains

All 200 individuals (50 per strain) represented unique multi-locus genotypes based on the microsatellite markers, specific characteristics of which are shown in Appendix A: Across strains, 7.9 (±2.9) alleles per locus were detected, resulting in comparable numbers of total alleles, with largely diagnostic patterns of locus-specific private alleles for all strains. The overall genetic differentiation was substantial (*F*_ST_ = 0.273) and significant for all pairwise comparisons between strains (*F*_ST_ ranging from 0.112 to 0.349) (Table 2). Across loci within populations, significant homozygote excess was found for strains S1 and S2 (Table 2), driven by significantly positive *F*_IS_ values of single loci for each strain (Appendix A). Locus-wide allelic richness (*A*_R_) and proportions of unique alleles (*A*_U_), i.e., mean numbers of different alleles across loci and overall proportions of alleles detected only once in 50 individuals per strain, were comparable across strains (Table 2). The estimated effective population sizes (*N*_e_) of the parental generation providing experimental larvae were in the range expected for such targeted designs with experimental subpopulations of moderate size, including synchronisation-related variation among strains for daily egg production (Table 2). Population genetic patterns described by the first three discriminant functions of DAPC are depicted in Figure 1A,B. Cluster analysis clearly detected four distinct genetic clusters (Figure 1C) corresponding to the four strains (Figure 1D). With reference to the global dataset of Kaya et al. [50], cluster assignment of experimental individuals was exclusive for S1 (cluster 2), S2 (cluster 6), and S3 (cluster 8), and almost exclusive for S4 (cluster 1) (Figure 1D).

### 3.2. Larval Development According to Genetic Background and Diet

Larval growth dynamics (mean live weights) of distinct BSF strains reared on different diets over time are depicted in Figure 2. All three main effects (strain: *F*_3,60_ = 337; diet: *F*_2,60_ = 6324; day: *F*_4,240_ = 26498), all twofold (strain × diet: *F*_6,60_ = 59; strain × day: *F*_12,240_ = 119; diet × day: *F*_8,240_ = 1499), and the threefold interactions (strain × diet × day: *F*_24,240_ = 19) were highly significant (*p* < 0.001). Harvest took place one day after the occurrence of the first prepupae and varied across combinations: on days 14 (S3 and S4) and 15 (S1 and S2) for poultry feed (PF), on day 16 (all strains) for food waste (FW), and on days 16 (S3 and S4) and 17 (S1 and S2) for poultry manure (PM). Despite comparable proportions of prepupae at harvest across diets, decreasing larval weights at the transition to the prepupal (6th) instar were solely observed on PF for all strains (Figure 2). Contrasts within a given factor across levels of the other factors are shown in Appendix A: in brief, (i) successive weight gains over time were significant, except between day 12 and harvest; (ii) generally, the highest larval weights were produced on PF, whereas larvae on PM remained lightest; (iii) strain S1 exhibited the highest and S2 the lowest overall mass gain across diets. However, given the strong interactive effects, formally correct threefold interaction contrasts had to be inspected for proper interpretation. Appendix A summarises pairwise comparisons between strains across pairs of diets within successive growth periods, as well as across the whole fattening period: concordant with numerous crossing lines in Figure 2; various significant threefold interactions were captured. Moreover, we studied the effects of BSF genetic background in-depth by pairwise contrasts between strains within fixed levels of diet and assessment days, as summarised in Appendix A; in brief, significant weight differences between strains were found on most days for almost all diets, with strikingly dynamic re-rankings of strains over time within and across diets.

For mean larval live weights at harvest, the effects of strain, diet, and their interaction were significant (Table 3, Figure 2 and Appendix A), including most twofold interaction contrasts being significant (Appendix A). Overall, larvae were heaviest on PF and lightest on PM. S1 and S2 ranked first and last, respectively, on all diets, whereas S3 and S4 were intermediate, with switched rankings on PM vs. PF and FW. While harvested larvae of S1 and S2 were heavier on PF than FW, this pattern was reversed for S3 and S4 (Table 3, Appendix A). Pairwise contrasts between strains within diets were all significant except for the comparison of S3 and S4 on PF (Table 3 and Appendix A).

Mortality was low overall, although significant effects of strain, diet, and their interaction were detected (Table 3, Appendix A). Few significant twofold interactions contrasts exclusively involved strain comparisons to S1 (Appendix A). Contrasts within dietary levels found no significant differences in mortality between strains on PM, but S1 survived significantly poorer than S3 and S4 on PF and poorer than all other strains on FW (Table 3, Appendix A).

### 3.3. Body Composition of Harvested Larvae: Concentrations and Total Amounts of Nutrients

The effects of strain, diet, and their interaction were significant for total larval biomass dry matter (DM), estimated protein (EP), ether extract (EE), and total ash (TA) (concentrations and amounts), as well as the EP:EE ratio (Table 3).

For BSFL total biomass (g DM), despite modest overall productivity on PM (Table 3), variable rankings of strains within diets mediated numerous significant twofold interactions (Appendix A). Contrasts between strains within diets showed that S1 was significantly most productive on all diets and that on FW, all strains varied significantly (Table 3, Appendix A).

The concentrations of larval EP (g/kg DM) were generally highest on PM and lowest on FW (Table 3, Appendix A), exhibiting some significant twofold interaction contrasts (Appendix A). The contrasts between strains within diets revealed EP concentrations being lowest for S1 on PF and FW, being also significantly lower than for S3 and S4 on PM (Table 3, Appendix A). While EP concentration of S2 was significantly higher than for S3 on PF and FW, the reverse was observed on PM. The EP concentration of S4 was significantly higher compared to S3 on FW and compared to S2 on PM. The total amounts of EP (g DM) showed partly divergent patterns (Table 3, Appendix A), as expressed by more prevalent significant twofold interactions (Appendix A). Contrasts between strains within diets identified significantly higher amounts of EP for S1 compared to all other strains on PF and PM (Table 3, Appendix A). While S2 produced significantly higher amounts of EP than S3 on PF, their ranking was reversed on PM. On FW, S2 ranked lowest for total EP yields.

Concentrations of larval EE (g/kg DM) varied remarkably across treatment combinations (Table 3, Appendix A), with most twofold interactions being significant (Appendix A). Generally, EE concentrations were highest on FW and lowest on PM (Table 3). Strain-specific pairwise contrasts within diets revealed that the EE concentration of S2 was lowest on FW, significantly lower compared to S3 and S4 on PF, but highest on PM (Table 3, Appendix A). Strain S1 featured an EE concentration of 50% DM on FW, a level that has so-far not been reported elsewhere. Slightly divergent patterns were observed for total amounts of EE (g DM), including fewer significant twofold interactions (Appendix A). Pairwise contrasts between strains within diets revealed significantly lowest amounts of EE for S2 on PF and FW and for S4 on PM, whereas S1 ranked significantly highest on FW (Table 3, Appendix A). 

The ratios of EP:EE varied substantially across diets (Table 3) and were characterised by numerous significant twofold interaction contrasts (Appendix A), as indicated by variable strain rankings (Appendix A). Pairwise contrasts between strains showed that EP:EE ratios were significantly lower for S2 than for S3 and S4 on PF, significantly lowest for S1 compared to all other strains on FW, and significantly lower for S4 than S2 and S3 on PM (Table 3, Appendix A).

For larval TA concentrations (g/kg DM), strong diet effects were detected, with values on PM being 2 and 7.5 times higher on average than on PF and FW, respectively (Table 3, Appendix A). Nevertheless, numerous significant twofold interaction contrasts (Appendix A), followed by pairwise contrasts within diets, indicated significant differences between strains; e.g., S1 ranked highest for TA concentration on PF but lowest on FW, with changing rankings of the other strains (Table 3, Appendix A). Similarly, the total amount of larval ash (g DM) also showed extensive twofold interactions (Appendix A), with highly variable rankings of strains across diets reflected in numerous significant pairwise contrasts (Table 3, Appendix A).

### 3.4. Larval Bioconversion Efficiency, Dietary Substrate Reduction Rates, and Systemic Losses

Significant effects of strain, diet, and their interaction were detected for BSFL bioconversion efficiency (BE) (Table 3, Appendix A). Strong differences between diets were found (up to a factor of 4), being highest for PF and lowest for PM (Table 3). Despite S1 exhibiting superior BE on all diets, differential performances of the other strains across diets resulted in significant pairwise contrasts, particularly on FW (Table 3, Appendix A), and mediated numerous significant twofold interactions (Appendix A). The maximum differences between strains ranged between 1.8% and 9.6% on PM and PF (Table 3). Yet, while S1 and S2 exhibited far lower BE on FW than on PF (~12% and 9%, respectively), differences between these two diets were less pronounced for S3 and S4 (~5% lower on FW).

Similarly, the effects of strain, diet, and their interaction were significant for proportional nitrogen utilisation (N-BE) of BSFL (Table 3). Albeit most twofold interaction contrasts were significant also for N-BE (Appendix A), their specific responses differed notably from those of BE, as corroborated by pairwise contrasts between strains within diets (Table 3, Appendix A). While S1 exhibited superior N-BE on PF and PM, on FW, S2 ranked significantly lower than all other strains, which did not differ. The N-BE of S2 was significantly lower than S3 and S4 on PM, but significantly higher than S3 and S4 on PF. 

Significant effects of strain, diet, and their interaction were found for larval reduction rate (RR) of dietary dry matter (DM) (Table 3, Appendix A). These were generally lowest on PM yet featured several significant twofold interactions (Appendix A). While S2 and S3 ranked lowest and second, respectively, for DM-RR on all diets, the highest-ranking strains were S1 on PF and S4 on both FW and PM (Table 3, Appendix A).

Significant main effects of strain and diet, but no interaction between them, were found for reduced rates of dietary neutral detergent fibre (NDF-RR) (Table 3), and accordingly, no significant twofold interactions (Appendix A). The effects of diet were striking, with NDF-RR being highest on FW and lowest on PM. NDF-RR did not differ between strains on PF and FW, but S3 and S4 differed significantly on PM (Table 3, Appendix A). Similarly, there were significant main effects on RR of acid detergent fibre (ADF), but no interaction (Table 3). Neither significant twofold interactions (Appendix A) nor pairwise contrasts between strains within diets (Appendix A) were detected for ADF-RR. For hemicellulose (HC) RR, however, significant effects of strain, diet, and their interaction were detected (Table 3), as well as some significant twofold interactions (Appendix A). Besides substantial diet effects, ranging from limited RR of HC on PM to almost complete degradation on FW (Table 3, Appendix A), pairwise contrasts between strains within diets revealed differences between S2 and S3 on FW, significantly lowest HC reduction of S4 on PM, but no differences among strains on PF (Table 3, Appendix A).

Significant effects of strain, diet, and their interaction were detected for proportional DM losses (Table 3), including several significant twofold interactions (Appendix A). Generally, DM losses on PF and PM ranged at similar levels, those on FW being substantially higher (Table 3, Appendix A). S1 and S2 lost the least DM on all diets and differed only on PF; by contrast, strains S3 and S4 lost significantly more DM, ranking highest on PF and FW, respectively (Table 3, Appendix A).

### 3.5. Amino Acid Profiles, Specific Protein Contents, and Nitrogen-to-Protein Conversion

The profiles of individual amino acids (AA; g/100 g BSFL protein), statistical evaluations of the main effects (strain and diet), and their interaction are summarised in Table 4, including abundances of significant twofold interaction contrasts plus pairwise contrasts between strains within diets and between diets within strains (for details see Appendix A). The profiles were generally affected by both main effects as well as by their interaction, however, showing interesting differences for specific AA responses (Table 4, Appendix A). Only for *Ile* no main nor interactive effects were significant. For the majority of essential AA, diet effects were strongest, but very clearly so only for *His* and *Lys*. Conversely, strain effects were more strongly pronounced for *Leu*, *Phe*, and *Thr* but still significant for all others (except *Ile*). Likewise, non-essential AAs were mostly characterised by significant main and interactive effects (Table 4). Exceptions were *Asx* (no diet effect), *Cys* (no strain effect), and *Ser* (no strain and interaction effects). Otherwise, strong diet effects were observed for *Arg*, *Glx,* and *Pro*, whereas strain effects were more pronounced or exceeding diet effects for *Ala*, *Gly,* and *Tyr*. While interaction effects were significant for all essential (except *Ile*) and non-essential AAs (except *Ser*), significant twofold interactions were more prevalent for non-essential AAs, apart from *Met* (Table 4, Appendix A).

Given the dynamic rankings of strains for AA profiles within diets (Table 4, Appendix A), only a few consistent patterns across diets could be inferred, such as the generally superior *Lys* contents of S2 or consistently lowest *His* and *Tyr* contents but highest *Ala* contents of S1. Conversely, in accordance with generally stronger diet effects, larval profiles of several AAs seem to reflect respective dietary profiles across all strains (Table 4 and Appendix A). For instance, positive correlations between larval and dietary AA profiles were indicated for *Lys*, *Phe,* and *Pro*, whereas they appeared to be negative for *Ala*. Relative larval contents of *Cys* and *Gly* tended to increase for all strains on PM, the diet containing the relatively highest levels thereof. Likewise, larval *Arg* contents tended to decrease for all strains on PM, the diet containing the least *Arg*. By contrast, the larval contents of *Glx* appeared to increase across strains on PM, the diet being the poorest regarding this AA. 

Specific protein (SP) concentrations (g/kg DM), as derived from analysed AA contents (including a proxy for *Trp*), yielded more accurate nitrogen-to-protein conversion factors (*Kp*) for BSFL of all strain-by-diet combinations (Table 4). Importantly, AA-based SP concentrations were consistently higher than EP concentrations derived from the general conversion factor of 4.76 proposed by Janssen et al. [76] (Table 3 and Table 4). Although statistical responses were largely congruent for EP and SP, the latter was characterised by stronger interactive effects (Table 4 and Appendix A). While on PF and FW, the ranking of strains was the same for EP and SP, albeit, with amplified relative differences for the latter, diverging strain rankings for EP and SP were found on PM (Table 3 and Table 4). Specifically, S2 and S3 were superior to S4 and S1, the latter showing the significantly lowest SP concentration on this diet (Table 4 and Appendix A). All strains featured relatively low SP concentrations on FW, though only for S3 all pairwise contrasts between diets were significant (Table 4 and Appendix A).

Lastly, larval *Kp* and, accordingly, non-protein-nitrogen (NPN) contents were also significantly influenced by diet and strain (Table 4, Appendix A). However, despite a marginally significant interaction between the main effects, none of the twofold interaction contrasts were significant (Table 4 and Appendix A). On PM, the diet containing by far the highest NPN contents, *Kp* was generally decreased, yet significantly higher for strains S2 and S3 than for S1 and S4. On PF, S3 was superior to S4, and on FW, no differences between strains were detected (Table 4 and Appendix A). Within each strain, *Kp* varied significantly between all diets, except for S2, which showed generally high *Kp* across diets (Table 4 and Appendix A).

## 4. Discussion

Our comprehensive experimental design of rearing larvae of four genetically distinct BSF strains on three different diets confirms previously reported overarching diet effects but crucially documents strong strain effects as well as ample genotype-by-diet interactions for virtually all investigated larval performance and body composition traits. These insights might explain part of the observed response variation among the numerous academic studies worldwide [4,5,6,33,42,43]. Therefore, in addition to recommended experimental standardisations [49], researchers are urged to genetically address managed or wild-sourced BSF strains appropriately (relative to the global diversity and population structure [48,50]) in future studies to facilitate purposive meta-analyses. Furthermore, considering the major role the BSF is supposed to play in the context of efforts towards a circular economy, improving agronomic sustainability as well as overall economic viability are both vital for the BSF farming sector. Our study stresses that matching the BSF’s genetic background to key environmental factors within any given setting, notably (but not exclusively) diet, is a so-far neglected aspect of crucial relevance for economic and ecological efficiency. Depending on specific purposes, applied potential comprises, e.g., tailored optimisation of relevant larval traits towards improved nutrient recycling during organic waste bioconversion or prioritising specific nutritional requirements of targeted fish or livestock recipients. Insofar, our key finding is that not any single one generally superior BSF strain can optimise all investigated performance traits as we demonstrate ample genetic variation across various traits of interest, as well as pronounced interactions between BSF strains and diets. This highlights a need for systematic phenotyping of worldwide BSF populations to illuminate genetic architectures of traits and their correlations, not least also to further progress dedicated selective breeding.

### 4.1. G × E Effects on Larval Fattening Performance and Body Composition Traits

When considering BSFL primarily as an alternative animal feed, total protein yield is a key parameter. Essentially, on any particular diet, certain BSF strains will be significantly more productive than others. However, total protein yield is only seemingly a trivial function of larval protein content and biomass productivity. We show that both contributing variables were similarly driven by G × E interactions and can thus compensate or reinforce effects on total protein yield. Moreover, associations between larval weights and either total BSFL biomass production or protein contents appear to change signs as well. This suggests that key downstream characteristics, such as protein efficiency, may be differentially receptive to improved breeding and will depend on actual combination-specific genetic correlations and the heritability of underlying traits (see below).

Besides quantitative protein yields, amino acid (AA) profiles could play a qualitative role in tailoring BSF meal inclusion levels in livestock and aquaculture feeding [3,87]. As previously documented [36,37,38,39], we here show that diet can significantly influence larval AA composition of any given strain, yet with notable variance among strains and generally less pronounced for essential compared to non-essential AA (differences of up to 0.84 g for the former and up to 3.07 g for the latter per 100 g protein). More relevant, our profiling revealed strain differences of up to 0.57 g for essential and up to 1.54 g for non-essential AAs (per 100 g protein) on any given diet. Causal mechanisms underlying the observed patterns may involve strain-specific (and diet-dependent) variations for utilising individual AAs, e.g., via compensatory metabolic pathways. Thereby components of developmental plasticity beyond the here-targeted readily accessible timing of shifting to the prepupal instar may trigger alterations of relative proportions of larval body tissues, particularly in the course of resource allocation conflicts regarding growth vs. fitness (see below), with measurable effects on overall protein efficiency. Such variation across strains appears economically relevant when extrapolated to industrial production scales. Apart from a prime role in optimally matching strain and diet, G × E interactions for most (including essential) AAs could also help mitigate the effects of particularly limiting dietary AAs through case-specific complementation of different resources (cf. [30,32]). Compared to generally limiting AAs relevant for conventional livestock, such as *Met*/*Cys* for poultry or *Lys* and *Thr* for pigs [79,88], primarily critical candidates for BSFL may differ (e.g., see [38,89,90]). In this regard, the finding that *Ile* was the sole AA for which no main or interactive effects were observed suggests that this candidate might represent an evolutionary conserved highly essential protein component for BSFL development.

Comparison of estimated protein (EP) concentrations derived from the recently proposed nitrogen-to-protein conversion factor (*Kp*) of 4.76 for whole BSFL [76] with specific protein (SP) concentrations based on analysed AAs revealed two aspects. First, intended as an adapted convention to facilitate generalisations for insects as feed and food, a *Kp* of 4.76 appears utmost conservative. Calculations for SP here align more closely with previously reported protein contents based on the classical conversion factor of 6.25 and subsequently corrected for chitin (e.g., [36]). Second, the pronounced strain effects, combined with only marginally significant interactions with diet detected for here-inferred *Kp* values, suggest there could be major limitations to such generalisations even within a single insect species. Thus, regardless of strong G×E effects on SP concentrations, patterns of specifically retrieved *Kp* imply similarly notable strain effects on larval non-protein-nitrogen (NPN) components. Chitin, constituting the dominant matrix of the BSFL exoskeleton (as opposed to sclerotin), is one such relevant NPN source [22]. However, the main contribution of chitin to larval body composition is carbohydrate [6,76], whereas other sources of carbohydrates in BSFL are controversially discussed [22] and likely negligible, such as muscle glycogen representing an essential resource for energy and lipid homoeostasis in insects [91]. Regarding overall mass balances, the expected general trend across diets of larval protein and fat contents being negatively correlated was notably absent for strain comparisons within diets. While a differentiated view on the role of BSFL fat content is discussed below, it appears that chitin content (carbohydrates plus NPN), previously reported to be affected by diet [36], is crucial to appropriately interpret observed differences in body composition mass balances between strains. Consequently, effects of BSF genetic background on generally important patterns of nitrogen-to-protein conversion as well as chitin contents (and composition) in particular, deserve further research, given the latter molecule may have both immune-stimulating as well as anti-nutritive effects in animals receiving BSFL meal [1,3,6,15,92].

### 4.2. G × E Effects on Bioconversion Efficiency and Sustainability Potential

A focus of BSFL applications on feed production prioritises bioconversion efficiency (BE), whereas organic waste treatment primarily targets dry matter (DM) reduction rates (RR) of larval diets. Yet, their interplay is inherently related to often neglected DM losses. Such systemic gaseous emissions are vital for sustainability evaluations [5,6,7], and knowledge of their nature and dynamics is accumulating only recently [93,94,95,96,97]. 

Strong differences for BE between diets (across strains) confirm previous studies using comparable diets [6,28,29,30,37]. Our study extends this finding to include significant BSF strain and interaction effects. As expected, BE positively correlated with total biomass, and both were largely related to nitrogen efficiency (N-BE). The latter exhibited pronounced strain and interactive effects as well, and, therefore, N-BE appears to be a major driver for BSFL productivity in general. Yet, productivity allows antagonistic interpretations, i.e., economic efficiency in terms of protein yields (producer’s perspective) vs. fitness maximisation via augmenting fat stores (insect’s perspective, see below).

Remarkably, we found neither BE nor DM-RR were precise predictors of DM losses. While both poultry feed (PF) and food waste (FW) fostered good larval growth, markedly higher DM-RR on the latter appear essentially attributable to increased emissions. Further, although DM losses on PF and poultry manure (PM) were numerically similar, generally lower BE on the latter diet identifies the relatively highest emission potential for rearing BSFL on PM. These data indicate a dilemma the BSF sector may face in the long term. While BSFL apparently show limited bioconversion efficiency coupled with greater DM losses on nutritionally poor diets, increased competition for high-quality feedstuff with other livestock should be avoided, as this could render the BSF an unnecessary trophic level. Favouring real waste exclusively exploitable by the BSF over feed grade material may hold the greatest sustainability potential from the wider nutrient-recycling perspective, at least relative to alternative options of treating organic waste [94,97], yet most likely at the cost of reduced insect farming productivity. Aiming particularly at recapturing nitrogen within the food chain, mixtures of animal manures with (feed-grade) materials that complement energy-rich nutrients could be promising to evaluate [30]. Irrespective of present regulatory hurdles regarding diets for insects for feed in some regions [2,5], BSF breeding programs are encouraged to not solely target production maximisation in a way that would somewhat thwart the basic concept of closing nutrient gaps and leaks but instead should prospectively focus on low-input strains complying with an overall sustainability valorisation. Substantial genotype and interactive effects for DM losses suggest such potential. For instance, differences of more than 3% DM losses between strains relative to BE hardly exceeding 8% for any strain on PM can be considered meaningful for life cycle assessments [5,6,7].

However, only a few studies so-far addressed BSF greenhouse gas emissions quantitatively [93,94,95,96,97], and even fewer included direct qualitative comparisons among diets [98,99] or BSF strains. Low N-BE coupled with substantial DM losses on PM suggests major emissions of nitrogen compounds. Conversely, high and more similar N-BE and larval protein concentrations on PF and FW, combined with increased fat contents and DM losses on the latter diet, suggest that high emission shares derive from cyclic CO_2_ generated during lipid metabolism [96,99]. Nevertheless, disentangling potentially variable contributions of metabolic pathways with a genetic basis vs. microbial processes to insect bioconversion efficiency and gaseous emissions remains notoriously difficult. Although diet can have only limited effects on altering BSFL core microbiota [100], most previous reports documented strong diet-related microbial dynamics [101,102,103,104]. Particularly population variation is interesting [105], and a recent study [106] supports that the G×E interactions for several relevant traits observed here may reflect even more complex interactions involving associated microbes. Identifying beneficial cultivable or transmittable microbial candidates depends on their characteristics and the nature of their host association. Specific microbial inoculations may positively influence production efficiency [107,108,109,110]. Yet, despite no clear evidence for vertical maternal imprinting for the time being [111], the BSF may possess such specific symbioses known from other insect systems (e.g., [112,113]), which thus may be amenable to fostering alternative inheritable solutions.

For instance, cellulose breakdown is likely microbe-mediated because, as for most insects, genes with cellulase activity are absent in the BSF genome [53], as corroborated by our finding that the degradation of NDF and ADF fractions were the only responses lacking G × E interactions. Compared to other responses, fibre reductions were characterised by greater variation across replicates within treatment combinations, indicating more stochastic microbial dynamics. Nevertheless, significant strain effects across diets suggest that host-microbe associations differ between genotypes in a surprisingly hardwired transgenerational manner. However, it remains to be explored to which extent variation in strain-associated symbionts can improve nutrient accessibility or digestibility of microbial biomass [111,114], as opposed to merely altering emission profiles without functional benefits for their host. In this context, the here observed interactions for hemicellulose reduction are interesting. Previously documented diet-mediated variation in BSFL midgut gene expression profiles [115] per se demonstrates that metabolic processes during digestion have a genetic basis. Hence, genetic variation across strains is not unexpected. Accordingly, the pervasive influence of an additional interaction level involving systemically relevant microbes exhibiting more or less tight host associations cannot be ruled out for the BSF [106].

### 4.3. G × E Effects in the Context of BSF Developmental Biology and Fitness

Since our experiment exclusively focussed on larval fattening, survival represented our most direct measure of fitness. Larval mortality is important from a production perspective for managing resource-demanding upstream egg and neonate production and keeping track of stocking density known to affect bioconversion efficiency [116,117,118]. Compared to previous reports [28,34,35,37], the here observed mortalities were very low overall yet revealed strong main and interactive effects. While no strain displayed mortalities higher than 0.7% on nutrient-poor PM, two strains showed significantly increased mortality of up to 5% on one or both nutrient-rich diets PF and FW. Environmentally plastic deleterious genetic variants may have escaped purging within populations before relevant dietary challenges occurred [119]. However, a putatively compromised digestibility of these strains could not be reconciled with their relatively higher growth performance on the same substrates. Instead, indirect effects of excess nutrients could have promoted critical microbial pathogens. Immune counter-defences, not only in insects, are generally considered costly and governed by trade-offs [114,120,121,122]. The BSF possesses a remarkable repertoire of immune genes [52], permitting complex response cascades [114,123,124], among which antimicrobial peptides are of transdisciplinary interest [6,125]. The latter were indeed previously shown to be alternatively expressed on different diets, including upregulation on protein and lipid-rich feed [126]. The precedence of growth at the cost of higher mortality vs. increasing survival at the cost of inferior growth performance may be differential outcomes across different genetic backgrounds, depending on the impact of conflicting resource allocation while combatting pathogens. Consistent with our results on PF and FW, this interpretation moreover coincides with our genetic data reflecting non-random survival only for the two strains that exhibited increased mortalities on nutrient-rich diets.

However, the here-observed interplay of exploiting dietary nitrogen resources for BSFL growth and the build-up of lipid stores allows additional inferences towards maximising fitness in a life-history context. Lipids are assumed to serve as the sole energy resource for the pupal and adult stages and are, therefore, expected to inherently translate into reproductive output [127], as feed uptake of adult flies is negligible [128]. Presumed associations between larval nutritional physiology and investment in fitness upon BSFL maturation (e.g., [33,41,129]) have only recently been evaluated by experimentally manipulating nutrients of artificial diets [130,131,132,133], including evidence that key metabolic responses to dietary conditions involve altered transcription profiles in BSFL fat bodies with relevance for protein and lipid metabolism [134]. Accordingly, during the transition from the fifth to the sixth (prepupal) larval instar, considered to be the critical phase for lipid metabolism, synthesis, and accumulation [41,135,136,137], particular signalling cascades may be under nuanced genetic control. Importantly, we observed that, across treatment combinations, N-BE appears to be a rather poor predictor of BSFL body weights at harvest, and the same applies to fat contents, which puts a previous study investigating one single strain on different diets into perspective [40]. Specifically, despite lower BE of strains S3 and S4 on FW relative to PF, similar N-BE on both diets permitted superior growth on FW, and both strains reached the highest fat contents on the diet on which they grew best. Conversely, strains S1 and S2 exhibited the highest N-BE and grew largest on PF but still reached their highest fat contents on FW. Moreover, relating N-BE to strain-specific growth slopes over time (Figure 2) suggests that the timing of critical physiological shifts varies genetically across strains and seems to generally occur well before the transition to the last larval instar. This aspect would thus still be important if industrial harvesting was scheduled before reaching the prepupal phase.

By nature, the BSF exploits nutritionally variable resources [138], yet a differentiated view of the huge dietary plasticity commonly observed in decomposing insects [139,140] is worthwhile. Our data suggest genetic variation in the BSF governing developmental cascades of lipid accumulation at the expense of further growth characterised by continued nitrogen metabolism. The variable G×E interactions for several metabolic responses found here, therefore, point towards strain-specific and diet-dependent fitness maximisation strategies rather than inadvertent (i.e., non-adaptive) pleiotropic effects. Previously neglected BSF genetic effects likely resulted in too simplistic concepts about general conspecific plasticity. Hence, apparently, more complex expressions thereof must have implications for defining and pursuing future breeding goals (next section).

Further applied relevance can be exemplified by two extreme scenarios. If the goal is to cope with seasonally or permanently diverse feedstuff, strains that exhibit increased plasticity across diets (while eventually lacking superior traits on any) are recommended. This is because their initial genetic makeup might differentially equip strains for adapting to new or changing diets, features not necessarily associated with genetic cluster assignment or patterns of overall diversity. On the other hand, such general-purpose strains may constantly perform sub-optimally in settings providing entirely standardised conditions. More detailed knowledge on G×E interactions linking nutritional physiology and developmental cascades would be particularly useful for highly controlled BSFL production settings so as to improve tailored feeding regimes for any BSF strain and avoid waste of resources (either in the form of temporary excess nutrients or deficiencies, which may trigger premature, irreversible developmental switches).

### 4.4. Implications for BSF Genetic Management and Breeding

The BSF strains investigated here merely represent 4 of the overall 16 distinct genetic clusters recently identified in a global population genetic survey [50]. Their rankings varied markedly across assessed traits, not permitting extrapolations beyond strains towards other global genetic clusters. The observed outcomes imply that the genetic makeup of the originally domesticated North American source harboured sufficient genetic variation to allow diversified breed formation and trait expression among distinct yet closely related genetic clusters 1 (here strain S4) and 2 (here strain S1) (see [50]). Such small-scale variation may well also be preserved across strains within clusters, which would limit cross-strain extrapolations even within specific clusters, but at the same time provide a reasonable basis for selective breeding largely independent of demographic history. Our findings further highlight that the genetic signature of BSF domestication, as previously defined by long-distance linkage characterising the globally most common managed strains of North American origin [50], does not imply commercial superiority per se. This, in turn, supports a stronger demographic rather than the breeding-mediated perspective of these unique genetic footprints of domestication [50]. Strains of naturalised Australasian clusters (strains S2 and S4 representing clusters 6 and 8, respectively; see [50]), supposedly characterised by a shorter breeding history in captivity, appear to be strong competitors in various regards. Accordingly, relatively quick breeding progress may be achieved even with other explicitly wild-sourced origins. At the same time, the breeding potential of strains deriving from the domesticated North American origin may be comparatively more limited due to severe prior demographic bottlenecks owing to its particular domestication history [50]. For instance, larvae of strains S2 and S3 exhibited fat contents comparable to strains S1 and S4 on PF and FW but higher lipid concentrations on PM, at the expense of body size. This suggests from an evolutionary perspective that non-domesticated BSF may not necessarily be better at digesting poor-nutrient diets but that they are superior in allocating resources to maintain high fitness on them. It would thus be interesting to test this hypothesis with truly wild-sourced strains from different regions. Although high fat contents are undesired for BSF meal production, selection advantages of fitness-related adaptation could be particularly expressed in semi-natural settings where complete BSF life cycles are implemented. 

This raises the principal question of whether selective breeding for BSF larval traits can be successful independent of their adult reproductive performance in the long term. As documented, e.g., in *Drosophila melanogaster* [141], holistic approaches that characterise multiple traits over entire life cycles, including BSF adult performance [142,143,144], are necessary to evaluate relevant genetic correlations or trade-offs, and to define indicator traits and selection goals of breeding programs. In this regard, decoupling the fattening and reproduction steps for economic or practical considerations deserves special attention [117]. On the one hand, spreading eggs or inoculation neonates from centralised breeding stations for stocking across decentralised fattening operations inevitably jeopardise optimisation in case they use different diets. On the other hand, maintaining an in-house breeding population on a diet that differs from that used for fattening might be justified from a hygienic perspective but may, in the worst case, represent a counter-selective scenario.

Future research will reveal the extent of phenotypic variation and the breeding potential of BSF strains from different genetic clusters [50]. Neutral genetic differentiation and structuring of worldwide wild populations, be they native or naturalised, may arguably reflect some signatures of diversified ecological adaptations [50], as commonly observed in arthropods [145,146]. Therefore, standardised phenotyping should not only consider prevalent mass-reared captive BSF strains, most of which are closely related. Screening trait variation across highly diverse locally resident wild populations all over different biogeographic regions, many of which are not yet subject to academic or commercial mass-rearing [48,50], is recommended to fully exploit the genetic potential of the BSF [51,119]. Accounting for ecologically relevant local adaptations would be particularly important for barely automated, semi-open farms of smallholders in the global south. They probably hold the greatest sustainability potential [1,12,147], but are also expected to be exposed to very specific and seasonally fluctuating environmental factors not limited to diet. Strain-mediated interactions could consequently be amplified. In such contexts, rather than introducing non-acclimatised BSF strains propagated elsewhere, it appears most reasonable to initiate managed strains with local native or naturalised BSF populations to better ensure long term resilience of such endeavours [116,148] (but see also [149]). Thereby, problems of introgression emanating from alien (domesticated) BSF strains into local wild populations could be mitigated alike [50,150,151]. By contrast, entrepreneurs maintaining escape-proof and highly automated large-scale productions with stable diets would likely benefit from choosing the most appropriate BSF strain matching specific artificial conditions, irrespective of its origin.

More generally, building future expansions of global phenotyping efforts on the recently published genomic characterisation of the BSF [52,53] renders many options for designing breeding schemes with tailored goals. Actors will be enabled to disentangle genetic architectures of traits, identify quantitative trait loci to obtain estimates of heritability, and investigate possible correlations among specific traits in various environments, as well as among traits across environments, to properly account for expected trade-offs relevant to performance [44,51]. Pervasive re-rankings of strains across performance traits, including changing signs of their associations, found here suggest that trait relationships change across environments. This has implications for choosing appropriate indicator traits and largely prevents translating selection progress across environments, which overall increases the anticipated complexity of breeding programs. However, we stress that breeding efforts should not be exclusively limited toward excessive performance maximisation within specific environments. Although the observed patterns suggest tailored breeding for the best progress, opportunities to select resilient strains from the beginning should not be missed. Such general-purpose genotypes [119] characterised by reasonable average performances for various traits, coupled with low variances across diets, may still be discovered, though unlikely to be found among the already pre-selected managed lineages of this genetically diverse insect. 

For the time being, however, without knowledge of the genetic architecture of any trait of interest, the common practice in BSF rearing facilities to interbreed various origins in a rather uncontrolled manner [50] is likely sub-optimal for achieving breeding progress. Despite the documented impact of intra-specific admixture for shaping global genetic patterns during contemporary non-native BSF range expansions, positive effects on commercially desired traits, collectively referred to as heterosis in other livestock, may turn out more complex in an insect featuring nuclear and mitochondrial genetic distances that effectively suggest a cryptic species complex [48,50].

## 5. Conclusions

Our new insights substantially contribute to the knowledge base of how to quantitatively and qualitatively influence larval production by paying more attention to the choice of BSF genetic background. Based on the rearing of four genetically distinct BSF strains on three diets with different nutrient composition we document for the first time that pronounced patterns of strain-by-diet interactions are widespread across investigated larval performance and body composition traits. This indicates that BSF genetic variation for any trait of interest matters when extrapolated to large-scale production. Therefore, the implications of matching BSF strains to specific dietary preconditions are of considerable economic and ecological relevance. Stronger efforts towards systematic screening of genotype-phenotype associations in BSF research are recommended to explore the genetic bases of key traits, together with their inventory of relevant variation, and make the best use of the genetic potential of this insect in an agricultural context. Accordingly, diversified selective breeding has huge potential and should emerge as a novel branch to support the growing BSF farming sector. However, besides tailoring breeds to various purposes, caution is advised not to define improvement solely based on the maximised performance of specific traits within too constrained (qualitatively high) environments. Resilient multi-purpose breeds that reliably perform across variable and particularly nutrient-poor waste streams should also be targeted to facilitate harmonising profitability with the sector’s hallmark of sustainability. 

## Figures and Tables

**Figure 1 insects-13-00424-f001:**
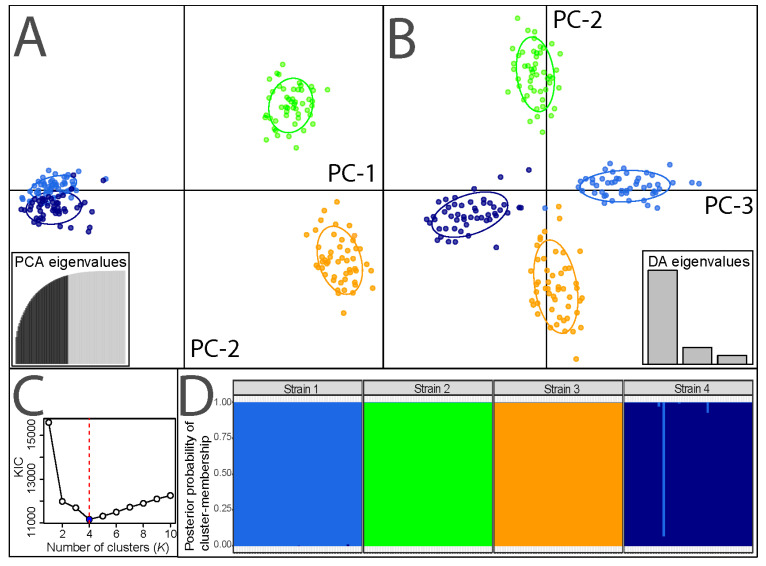
Population genetic patterns and cluster analysis of experimental black soldier fly strains. Discriminant analysis of principal components of 50 microsatellite-based multi-locus genotypes (dots) per strain (S1: light blue; S2: green; S3: orange; S4: dark blue): (**A**) axes 1 and 2; (**B**) axes 2 and 3. (**C**) Model-based inference of the optimal number of genetic clusters in the data set according to KIC goodness-of-fit statistics. (**D**) Posterior membership probabilities of individuals horizontally arranged as vertical bands (stacked bar plots) within strains (colours for inferred cluster assignments are harmonised according to the global population genetic survey by Kaya et al. [50], see Appendix A).

**Figure 2 insects-13-00424-f002:**
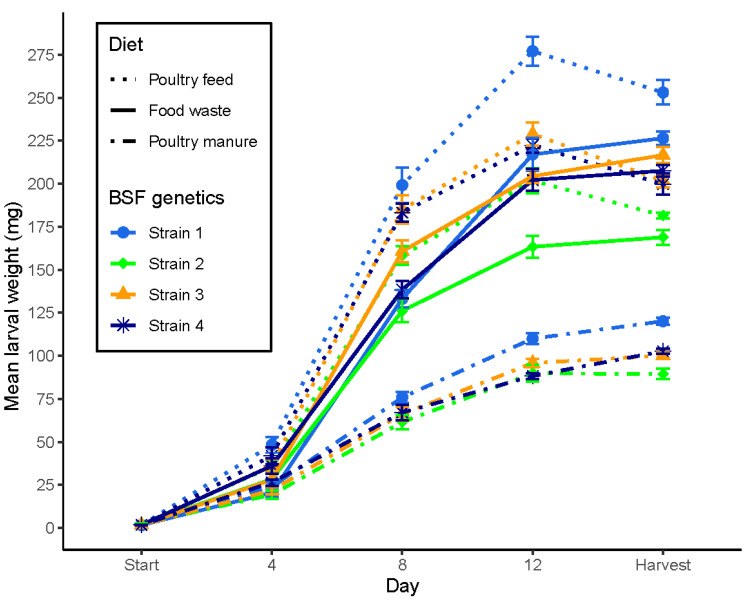
Larval growth dynamics of individual black soldier fly strains on different diets over time. Colours identifying strains match those in Figure 1, and error bars depict standard deviations across replicates. Depending on the occurrence of the first prepupae, harvest dates for individual combinations varied between days 14 and 17 (see main text). For a statistical summary, see Appendix A.

**Table 1 insects-13-00424-t001:** Chemical composition (g/kg dry matter) of experimental diets. Total dry matter (DM) of provided rations per replicate (1100 larvae each) were 312 g (PF), 432 g (FW), and 399 g (PM).

Item	Poultry Feed (PF)	Food Waste (FW)	Poultry Manure (PM)
Total ash	132.4	35.3	257.5
Carbon	440.5	468.2	381.0
Nitrogen	32.3	24.5	46.4
Ether extract	50.0	52.2	23.0
Neutral detergent fibre	299.1	317.3	399.3
Acid-detergent fibre	132.2	102.1	298.6
Hemicellulose	166.9	215.2	100.7

**Table 2 insects-13-00424-t002:** Strain-specific population genetic characteristics and pairwise genetic differentiation of experimental black soldier fly strains. *F*_ST_: fixation index indicating pairwise genetic differentiation among strains (S1–4); *F*_IS_: inbreeding coefficient; *H*_obs_/*H*_exp_: observed and expected heterozygosity; *A*_R_: mean allelic richness across loci; *A*_U_: proportion of unique alleles (detected only once); *N*_e_: estimated effective population size (parental generation). Significant *F*-statistics are highlighted in bold (*p* < 0.001). Patterns refer to cross-locus comparisons; for evaluations of individual microsatellite loci see Appendix A, and for mitochondrial COI haplotype distance matrix, see Appendix A.

Strain	Pairwise *F*_ST_			*F* _IS_	*H* _obs_	*H* _exp_	*A* _R_	*A* _U_	*N_e_*
	S1	S2	S3						
S1				**0.129**	0.496	0.563	4.000	0.108	110
S2	**0.330**			**0.135**	0.432	0.494	3.867	0.074	193
S3	**0.307**	**0.112**		0.021	0.551	0.557	4.467	0.114	267
S4	**0.132**	**0.349**	**0.335**	−0.013	0.528	0.516	3.667	0.066	147

**Table 3 insects-13-00424-t003:** Larval performance and compositional data of four black soldier fly strains reared on three diets. DM: dry matter; N: nitrogen. Applied models included weighted linear regression based on log-transformed (i) and arcsine square root-transformed (ii) responses, generalised linear models fitted with logit link and binomial (iii) or quasi-binomial probability distribution (iv). Effect sizing and testing refer to respective *F* or *χ^2^* statistics. For the main effects of strain (S), diet (D), and their interaction (60 residual degrees of freedom (DF) throughout), asterisks indicate statistical significance (***: *p* < 0.001). Italic numbers in parentheses associated with interaction terms denote numbers of significant twofold interaction contrasts (out of 18 tests throughout), see Appendix A. Superscripts for strain-specific responses denote significant pairwise contrasts within fixed dietary levels (^a-d^: PF; ^e-h^: FW; ^i-l^: PM), see Appendix A. For the nutritional composition of diets, see Table 1. Appendix A show boxplots specifying replicate-wide scattering (medians, 25–75% percentiles and whiskers) plus standard errors of the here presented means across six replicates within factorial combinations for all of the listed parameters.

					Diet											
		Statistics			Poultry Feed (PF)				Food Waste (FW)				Poultry Manure (PM)			
Parameter	Model; Test	Strain (DF = 3)	Diet (DF = 2)	S × D (DF = 6)	Strain S1	Strain S2	Strain S3	Strain S4	Strain S1	Strain S2	Strain S3	Strain S4	Strain S1	Strain S2	Strain S3	Strain S4
Larval live weight (mg) at harvest	i; *F:*	634.2 ***	10497.2 ***	27.3 *** (*10*)	253.0 ^a^	181.5 ^c^	201.8 ^b^	199.7 ^b^	226.4 ^e^	168.9 ^h^	216.6 ^f^	207.5 ^g^	120.1 ^i^	89.4 ^l^	100.0 ^k^	102.6 ^j^
Larval mortality (%)	ii; *F:*	41.1 ***	15.9 ***	6.6 *** (*3*)	5.14 ^a^	2.03 ^ab^	0.67 ^b^	0.65 ^b^	5.00 ^e^	0.64^f^	0.20^f^	0.83^f^	0.41	0.70	0.36	0.24
Larval biomass (g DM)	i; *F:*	848.6 ***	17312.8 ***	29.7 *** (*11*)	107.5 ^a^	80.7 ^b^	78.0 ^b^	77.5 ^b^	97.6 ^e^	72.4 ^h^	88.6 ^f^	83.3 ^g^	34.0 ^i^	26.9 ^j^	27.9 ^j^	27.5 ^j^
Estimated protein (EP; g N × 4.76/kg larval DM)	iv; *χ^2^:*	135.5 ***	1037.5 ***	42.3 *** (*6*)	303 ^c^	322 ^a^	313 ^b^	319 ^ab^	272 ^g^	300 ^e^	290 ^f^	307 ^e^	339 ^j^	347 ^j^	358 ^i^	356 ^i^
Total estimated protein (EP; g)	i; *F:*	505.0 ***	21,638.8 ***	37.6 *** (*12*)	32.6 ^a^	26.0 ^b^	24.4 ^c^	24.7 ^bc^	26.5 ^e^	21.7 ^f^	25.6 ^e^	25.5 ^e^	11.5 ^i^	9.3 ^k^	10.0 ^j^	9.8 ^jk^
Ether extract (EE; g/kg larval DM)	iv, *χ^2^:*	20.7 ***	1439.7 ***	101.7 *** (*11*)	279 ^b^	232 ^b^	381 ^a^	395 ^a^	500 ^e^	441 ^f^	454 ^ef^	485 ^ef^	78 ^j^	140 ^i^	90 ^j^	71 ^j^
Total ether extract (EE; g)	i; *F:*	99.1 ***	6925.8 ***	8.8 *** (*8*)	29.7 ^a^	18.7 ^b^	29.7 ^a^	30.7 ^a^	48.8 ^e^	31.9 ^g^	40.2 ^f^	40.4 ^f^	2.7 ^i^	3.8 ^i^	2.5 ^i^	2.0 ^j^
Estimated protein: ether extract ratio (EP:EE)	i; *F:*	686.3 ***	3432.5 ***	14.5 *** (*13*)	1.13 ^ab^	1.41 ^a^	0.82 ^b^	0.81 ^b^	0.55 ^f^	0.68 ^e^	0.64 ^e^	0.63 ^e^	4.38 ^ij^	2.66 ^j^	3.97 ^j^	5.06 ^i^
Total ash (TA; g/kg larval DM)	iii; *χ^2^:*	59.5 ***	43,845.0 ***	172.4 *** (12)	112 ^a^	101 ^b^^c^	102 ^b^	9.7 ^c^	24 ^f^	28 ^e^	30 ^e^	31 ^e^	213 ^k^	201 ^l^	217 ^jk^	223 ^i^
Total ash (TA; g)	i; *F:*	912.3 ***	10372.4 ***	48.9 *** (15)	12.0 ^a^	8.2 ^b^	8.0 ^b^	7.6 ^b^	2.3 ^g^	2.1 ^h^	2.7 ^e^	2.6 ^f^	7.3 ^i^	5.4 ^k^	6.1 ^j^	6.2 ^j^
Bioconversion efficiency (BE; g larvae/kg diet; DM-based)	ii; *F:*	990.3 ***	18997.2 ***	42.7 *** (12)	343 ^a^	256 ^b^	248 ^b^	247 ^b^	225 ^e^	166 ^h^	204 ^f^	192 ^g^	84 ^i^	66 ^j^	69 ^j^	68 ^j^
Nitrogen efficiency (N-BE; g N larvae/kg N diet)	ii; *F:*	2510.0 ***	42850.3 ***	58.5 *** (12)	674 ^a^	535 ^b^	503 ^c^	510 ^c^	522 ^e^	424 ^f^	505 ^e^	488 ^e^	128 ^i^	102 ^k^	111 ^j^	109 ^j^
Dry matter (DM) reduction (g/kg diet)	ii; *F:*	5807.6 ***	23536.8 ***	13.4 *** (10)	638 ^a^	527 ^c^	631 ^a^	600 ^b^	771 ^g^	697 ^h^	786 ^f^	802 ^e^	359 ^j^	344 ^k^	366 ^j^	375 ^i^
Neutral detergent fibre (NDF) reduction (g/kg diet)	ii; *F:*	533.7 ***	2118.8 ***	1.5 (0)	593	598	599	585	833	814	840	832	189 ^ij^	175 ^ij^	233 ^i^	166 ^j^
Acid detergent fibre (ADF) reduction (g/kg diet)	ii; *F:*	48.9 ***	368.6 ***	0.9 (0)	479	456	499	461	531	525	552	587	146	154	215	179
Hemicellulose (HC) reduction (g/kg diet)	ii; *F:*	255.0 ***	2469.0 ***	11.6 *** (8)	684	710	678	683	976 ^ef^	952 ^f^	977 ^e^	960 ^ef^	316 ^i^	237 ^i^	285 ^i^	128 ^j^
Dry matter (DM) losses (g emissions/kg diet)	iv, *χ^2^:*	297.7 ***	5986.1 ***	97.0 *** (7)	294 ^c^	270 ^d^	381 ^a^	353 ^b^	546 ^g^	545 ^g^	582 ^f^	610 ^e^	274 ^k^	278 ^jk^	297 ^ij^	307 ^i^

**Table 4 insects-13-00424-t004:** Profiles of larval amino acids, specific protein concentration, and nitrogen-to-protein conversion. BSFL: black soldier fly larvae. Abbreviations for amino acids (AA) refer to conventional nomenclature. Concentrations (g/100 g protein) of essential AAs are listed first, followed by non-essential AAs. Specific protein concentrations based on summed amino acid contents (g/kg dry matter (DM)) as well as nitrogen-to-protein conversion factors (*Kp*; g protein/g nitrogen) were calculated, accounting for the general literature-derived proxies for *Trp* applied to larvae from all combinations and for all diets, i.e., 1.8 and 1.2 g/100 g protein, respectively. Effects of the factors strain (S), diet (D), and their interaction are indicated by *F*-values of linear models (^1^ denotes responses assessed by weighted regression). Significance levels (60 residual degrees of freedom (DF) throughout) are shown (*: *p* < 0.05; ***: *p* < 0.001). Italic numbers in parentheses associated with interaction terms specify numbers of significant twofold interaction contrasts (out of 18 tests throughout), see Appendix A. Values for diets are shown (first column in each vertical diet block) but are not subject to statistical evaluations. Small superscript letters denote significant contrasts (*p* < 0.05) between strains (S1–4) within dietary levels (i.e., rows within column blocks for specific diets: ^a–d^: PF; ^e–h^: FW; ^i–l^: PM), see Appendix A. Capital superscript letters denote significant contrasts between dietary treatments within fixed levels of strain (i.e., rows within strain-specific columns across vertical diet blocks: ^A–C^: S1; ^D–F^: S2; ^G–I^: S3; ^J–L^: S4), see Appendix A. Appendix A specify replicate-wide standard deviations within factorial combinations for all listed parameters.

Amino Acid (g/100 g protein)	Effect			Poultry Feed (PF)					Food Waste (FW)					Poultry Manure (PM)				
					BSFL					BSFL					BSFL			
	Strain (DF = 3)	Diet (DF = 2)	S × D (DF = 6)		Strain S1	Strain S2	Strain S3	Strain S4		Strain S1	Strain S2	Strain S3	Strain S4		Strain S1	Strain S2	Strain S3	Strain S4
Essential																		
*His* ^1^	13.72 ***	321.47 ***	4.51 *** (*2*)	2.61	3.37 ^c A^	3.45 ^c D^	3.53 ^b G^	3.65 ^a J^	2.62	3.37 ^g A^	3.46 ^f D^	3.43 ^fg H^	3.57 ^e J^	2.09	2.90 ^j B^	3.14 ^i E^	3.03 ^ij I^	3.10 ^i K^
*Ile* ^1^	0.72	1.92	0.74	4.10	4.57	4.53	4.57	4.60 ^JK^	4.18	4.52	4.50	4.58	4.47 ^K^	4.68	4.57	4.55	4.58	4.74 ^J^
*Leu* ^1^	12.80 ***	3.40 *	2.46 * (*1*)	9.21	7.67 ^a B^	7.54 ^b^	7.60 ^ab^	7.71 ^ab J^	7.46	7.88 ^e A^	7.63 ^f^	7.64 ^f^	7.56 ^f JK^	8.10	7.65 ^i B^	7.58 ^ij^	7.56 ^ij^	7.49 ^j K^
*Lys*	32.95 ***	174.45 ***	6.68 *** (*4*)	5.06	6.62 ^b B^	7.07 ^a E^	7.04 ^a H^	6.81 ^b K^	4.28	6.81 ^f B^	7.19 ^e E^	6.62 ^f I^	6.63 ^f K^	5.83	7.35 ^j A^	7.76 ^i D^	7.40 ^j G^	7.47 ^j J^
*Met* ^1^	3.53 *	13.19 ***	18.19 *** (*11*)	2.39	2.05 ^b B^	2.19 ^a D^	2.11 ^b G^	2.13 ^ab^	2.27	2.28 ^e A^	2.15 ^f D^	1.95 ^g H^	2.10 ^f^	1.85	1.95 ^k B^	2.04 ^jk E^	2.12 ^ij G^	2.15 ^i^
*Phe* ^1^	42.64 ***	68.75 ***	7.30 *** (*5*)	5.45	4.92 ^b A^	5.15 ^a D^	4.91 ^b G^	4.89 ^b J^	5.31	4.87 ^ef A^	4.89 ^e E^	4.72 ^fg H^	4.72 ^g K^	4.44	4.69 ^j B^	4.77 ^ij F^	4.84 ^ij GH^	4.86 ^i J^
*Thr* ^1^	58.51 ***	33.09 ***	8.32 *** (*4*)	3.73	4.08 ^B^	4.07 ^D^	4.06 ^H^	4.06 ^K^	3.59	4.30 ^A^	4.19 ^D^	4.20 ^G^	4.18 ^JK^	5.21	4.09 ^j B^	3.97 ^k E^	4.15 ^ij GH^	4.26 ^i J^
*Val* ^1^	8.36 ***	15.30 ***	4.91 *** (*3*)	4.78	6.22 ^c^	6.47 ^b D^	6.71 ^a G^	6.59 ^a J^	4.99	6.45 ^ef^	6.13 ^f E^	6.40 ^ef H^	6.55 ^e JK^	6.28	6.21	6.46 ^D^	6.27 ^H^	6.36 ^K^
Non-essential																		
*Ala* ^1^	580.40 ***	142.26 ***	16.57 *** (*9*)	5.67	8.29 ^a B^	7.75 ^b E^	7.22 ^c H^	7.20 ^c K^	4.37	9.34 ^e A^	8.12 ^f D^	8.12 ^f G^	7.80 ^g J^	9.69	7.95 ^i B^	7.46 ^j F^	7.36 ^j H^	7.58 ^j J^
*Arg* ^1^	63.38 ***	425.00 ***	48.50 *** (*15*)	6.64	5.39 ^bc A^	5.37 ^c E^	5.57 ^ab G^	5.65 ^a J^	5.76	4.65 ^g B^	5.77 ^e D^	5.59 ^f G^	5.70 ^ef J^	4.38	4.81 ^j B^	5.16 ^i F^	4.90 ^j H^	4.77 ^j K^
*Asx*	18.71 ***	2.01	9.38 *** (*4*)	8.17	9.17 ^a^	9.14 ^a D^	8.75 ^b G^	8.74 ^b^	6.34	9.06 ^e^	8.64 ^f E^	9.02 ^e H^	8.67 ^f^	9.25	9.00 ^i^	8.59 ^j E^	9.28 ^i H^	8.59 ^j^
*Cys*	0.97	832.00 ***	11.77 *** (*9*)	1.95	0.96 ^a B^	0.98 ^a E^	1.00 ^a H^	0.90 ^b L^	2.56	0.97 ^B^	1.00 ^E^	0.99 ^H^	0.97 ^K^	2.70	1.36 ^i A^	1.29 ^j D^	1.27 ^j G^	1.37 ^i J^
*Glx* ^1^	198.53 ***	784.70 ***	13.53 *** (*9*)	19.63	11.26 ^B^	11.18 ^E^	11.15 ^H^	11.11 ^L^	23.57	10.31 ^g C^	11.33 ^ef E^	11.16 ^f H^	11.42 ^e K^	13.68	13.38 ^i A^	12.76 ^jk D^	12.52 ^k G^	13.10 ^j L^
*Gly* ^1^	37.30 ***	65.37 ***	10.29 *** (*6*)	4.39	5.54 ^a B^	5.39 ^b E^	5.41 ^b H^	5.59 ^a K^	3.75	5.87 ^e A^	5.44 ^f E^	5.48 ^f H^	5.45 ^f L^	6.95	5.90 ^i A^	5.66 ^k D^	5.69 ^jk G^	5.86 ^ij J^
*Pro* ^1^	24.52 ***	87.01 ***	5.43 *** (*3*)	7.23	6.14 ^B^	6.13 ^DE^	6.07 ^H^	6.48 ^J^	9.73	6.90 ^e A^	6.27 ^f D^	6.68 ^e G^	6.54 ^ef J^	5.48	5.69 ^ij C^	5.88 ^i E^	5.85 ^i H^	5.51 ^j K^
*Ser*	1.94	11.45 ***	1.05	4.98	4.34	4.13	4.20 ^H^	4.10 ^K^	5.75	4.42	4.37	4.46 ^G^	4.52 ^J^	5.19	4.46	4.30	4.34 ^GH^	4.29 ^JK^
*Tyr* ^1^	219.90 ***	293.08 ***	10.83 *** (*6*)	2.89	7.75 ^b A^	7.82 ^b D^	8.41 ^a G^	8.18 ^a J^	2.36	6.58 ^f B^	7.33 ^e E^	7.30 ^e H^	7.48 ^e K^	3.10	6.51 ^k B^	7.02 ^j F^	7.19 ^i H^	6.86 ^j L^
Specific protein (g/kg DM) ^1^	26.40 ***	230.63 ***	11.83 *** (*12*)	196.4	362.6 ^c A^	384.2 ^a D^	373.8 ^b H^	375.7 ^b J^	139.8	311.3 ^g B^	351.2 ^e E^	338.4 ^f I^	355.0 ^e K^	93.9	360.3 ^k A^	383.8 ^i D^	389.4 ^i G^	372.0 ^j J^
*Kp* (×N) ^1^	23.77 ***	360.48 ***	2.52 * (*0*)	6.08	5.70 ^ab A^	5.68 ^ab D^	5.69 ^a G^	5.61 ^b J^	5.70	5.46 ^B^	5.58 ^D^	5.56 ^H^	5.51 ^K^	2.02	5.07 ^j C^	5.28 ^i E^	5.18 ^i I^	4.97 ^j L^

## Data Availability

All data generated or analysed during this study are included or referred to appropriately in this article and its supplementary information file.

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
