# Peer review of "Genotype-by-Diet Interactions for Larval Performance and Body Composition Traits in the Black Soldier Fly, Hermetia illucens"

_insects, 2022, doi:10.3390/insects13050424_

Round 1

Reviewer 1 Report

Dear authors

Following are some of the comments

1) Use of English language needs to improve. As there are many spell errors - for instance, in 2nd para of introduction section evidence is spelled as evince.

2) Introduction section needs to be divided into five paragraphs. The first para explains what is the issue, 2nd para - why is it an issue, 3) what is in the past or currently being done in order to resolve that issue and finally, structure of the paper. Use some latest references from the same journal  and reference the following article Codesign of food system and circular economy approaches for the development of livestock feeds from insect larvae

3) Methodology and results and discussion section reads well.

4) Conclusion section needs to show the novelty of this research more prominently. 

Overall, it's a good piece of research work.

Regards

Reviewer 2 Report

I liked this paper, with my low experience due to my young age as researcher, I have never found before a precise paper with access to the main results as you have done. I suggested changing on the results presenting to simplify the reading and the understanding for the readers. I would like to have more time to discuss the results with you.

Materials and method

Line 139-148: please to change the order of the strains, from S1 to S4. It results in clearer system of reading and understanding.

Line 151-164: please to insert description and of the genetics index before to find them in the results (table2). It could be clearer.

Line 192: Is it a reference or your personal observation with these three diets? Referring to “… containing equivalent of 77mg wet weight…”

Line 198: daily or batch feeding system? Why did you choose one of them?

Line 198-202: did you record also pH of initial three diets? If yes, please insert it is could be important.

Line 203-205: You can add estimate protein of the three diet and the coefficient Kp. Did you evakluate carbohydrate? They are really important considering BSF as reported by Cammack and Tomberlin, 2017 and Barragàn-Fonseca et al., 2019. Can you add, please?

Line 214: Did you set this humidity (50% RH) to allow reducing moisture during experiment and allowing to have an easy separable and sieving materials(frass) from larvae?

Line 228: Before freezing, did you gut 24hour the larvae before freezeing? It could impact on th AA composition analysis.

Line 230. It is clear, you can delete “individual”, you had put “of each replicate”.

Line 244-250: ADL? Did you analyse?

Line 248-254 Put the formulae in clearer view compared to this one.

Line 266: Which is Kp for poultry manure?

Line 269 to 301: Did you use contrast as post-hoc, right? Did you explain 6 replicate per treatment?

Line 269-302Why did you not referee to DAPC in statical paragraph? You talk about this only in the results.

Line 298: When you say A PRIORI, you mean only between strains and you do not consider difference between diets, right?

Line 305: 200, referee to 50 larvae per strains for analyzing for genetics indexes?

Line 309: Please, you should explain here in the text Fst and the meaning, I got later but it is not clear.

Line 311 What do you mean high? Specify a high value or explain a range: exemple from low, normal to high values.

Line 313: insert Ar and explain: allelic richness, it lacks at all.

Line 312 to line 321: Insert Hobs and Hexp comments on these result of Table 2.

Line 318: explain DAPC in statistical methods

Line 323 to line 330: you say here Au: proportion of unique alleles. But in the text, you say proportion of rare alleles, please uniform it.

Line 331: Fig. D: it is clear only for a specialist of this image, maybe a better.

Fig 1. C: Why did you say 4 cluster is the right results? I got it but it is not clear for other reader.

Line 339: if you say difference between diet was a “A PRIORI”(line 300) you could split in three graphs. Please Split in three figure the figure 2:; example: diet PF (Figure2), Diet FW (Figure 3) and Diet PM (Figure 4). You can explain interaction in the text.

Line 353 to line 371: It is very clear part.

Line 387: Please put the abbreviations in the table 3, after splitting it. (Example: larval mortality (LM), etc…)

Line 378 to line 491; Table 3: please split the table 3 in three table:

  1. Growth performances (larval development according to genetic background and diet)
  2. Bioconversion efficiency (Larval bioconversion efficiency, dietary substrate ….)
  3. Chemical analysis (Body composition of harvested larvae)

Now it is impossible in the following pages understand each paragraph. You can insert the table for each paragraph to make easier the reading and understanding for the reader. RENAME EACH TABLE FROM HERE

Line 386: did you use standard error or standard deviation. I understood you used in the boxplots and you did not explain in the table.

Line 422: very interesting point on 50% EE

Line 432: I think you wanted to write “lower” instead of “lowest”

Line 444: Split table 3 for Bioconversion efficiency (Larval bioconversion efficiency, dietary substrate ….)

Line 451: where did you read 2-10%? I read from 6.6% to 8.4% PM ans 16.6 to 22.5% in FW. I don’t understand what you are meaning. Are you talking about BE (Bioconversion efficiency), right?

Line 466: It is not correct, see table 3 for DM-RR.

If you split table 3 it would be easier to comment for you and understanding for the reader.

Line 494: rename the table.

Line 536: Are you meaning the strain influence the coefficient Kp of the protein too?

It is a really interesting point. It means in insect it would not so efficient as method to have the protein content. As can we solutionate it?

Line 554: Table4: to insert a name of these columns in the diet column, now it is without description in the table even if it is without statistical analysis

Methionine: in the table the value of Methionine is very high compared to other paper, it is usually around 0.6 values.

Line 586 to 644: I suggest moving this part later than Effect of bioconversion and development. I suggest to use the same order you used in the results.It would be clearer than the current order.

Line 662 to line 669: maybe the PM could be integrated in a mixed diet between this kind of waste with other rich in carbohydrate to obtain a more efficient diet for BSFL.The point you talked about is interesting for environmental point of view too.

Line 699 to line 709: the point about cellulose is very interesting. Maybe a fermentation operated by cellulolytic fungi can be impacted by BSF strains. Because of the growth of fungi could be eated by BSF larvae more than digested cellulose products.

Line 699 to line 715: you should think on a microbiota gut community and if they are connected to the genetics strains. This is a big question for the future. You could discuss a little bit this point, maybe with similar study on other Diptera of entomological interest.

Line 716 to 782: I would suggest to move to the first part of the discussion later the first paragraph (from line 585).

Line 716 to 725: The mortality is really similar to many other studies ranging between 2 and 5%.

Higher mortality is common with animal waste or by-product.

Line 728 to line 739: in the industry of BSF is more important the final biomass and dry matter you can obtain. If the mortality is higher but you obtain more DM at the end. It looks more profitable. This study is important because of it demonstrates the strains effect on the larval development and DM production. It is a starter point for the future analysis on the genetics for BSF industry.

Line 750 to 782: the point is very interesting, but you have also to consider the reality of the formulation of diet for BSF. We cannot choose too much the diet-component because of we need to pay less as possible the by-products to obtain a good economic balance, for instance we are producing a feed and not a food for now, in Europe and it should cost less is possible. To try to figure out about the nutrients requirements for BSF is important but we do not fix too much on that point is more important now is we have a strain adapted more for this waste or for another one. It change very much depending on the strain as you demonstrate in your research. This research is important for that. The BSF company did not very well which kind of genetics they have and how they work better. Moreover, it is important to discuss about the coevolution of BSF larvae to the diet we use for them. They coevolve with it if we maintain the diet fixed.

Line 806 to line 811: I think the point here is about genetics expression of adaptive genes to digest low value nutrient as fiber and they can transform it in fat. Moreover, wild BSF strain should be evolved on the poor material as manure or other decaying organic matter and then, they are not selected by humans for growing on other more accessible vegetable by-product.

The humans operated on the BSF genetics selection using not natural diet for BSF as cereal by-products.

The BSF strains with higher variability are the most adapted to low value material. It should be confirmed by a GxE on the other 12 genetics described by Ståls et al., 2020. and Kaya et al. (2021).

In general, this paper is very well done and complex at the same time. It is one of the first paer on the genetic diversity not only at gene point of view but it looks on the performances of BSFL.

The next stage would be on the adult I think eggs production and hatchability rate.
